# A 20-Year Retrospective Study of Children and Adolescents Treated by the Three-in-One Procedure for Patellar Realignment

**DOI:** 10.3390/jcm12020702

**Published:** 2023-01-16

**Authors:** Giovanni Trisolino, Alessandro Depaoli, Giovanni Gallone, Marco Ramella, Eleonora Olivotto, Paola Zarantonello, Stefano Stallone, Valentina Persiani, Giacomo Casadei, Gino Rocca

**Affiliations:** 1Pediatric Orthopedics and Traumatology, IRCSS—Istituto Ortopedico Rizzoli, 40136 Bologna, Italy; 2RAMSES Laboratory, RIT Department, IRCSS—Istituto Ortopedico Rizzoli, 40136 Bologna, Italy; 3Department of Trauma and Orthopedic Surgery, Maggiore Hospital, 40133 Bologna, Italy; 4Unit of Orthopedics and Traumatology, IRCCS—Policlinico Sant’Orsola-Malpighi, 40137 Bologna, Italy

**Keywords:** patellar instability, pediatric, three-in-one procedure, green procedure, Roux-Goldthwait, syndrome, congenital ligamentous laxity, neurologic, obligatory patella dislocation, fixed patella dislocation

## Abstract

Background: Patellar instability is the most common disorder of the knee during childhood and adolescence. Surgical treatment significantly reduces the rate of redislocation, but the underlying pathologies and pattern of instability may affect the results. We aimed to report the clinical and functional outcomes of the three-in-one procedure for patellar realignment in a cohort of skeletally immature patients with or without syndromes and various patterns of chronic patellar instability. Methods: We retrospectively investigated 126 skeletally immature patients (168 knees) affected by idiopathic or syndromic patellar instability, who underwent patella realignment through a three-in-one procedure. We classified the instability according to the score proposed by Parikh and Lykissas. Results: Patellar dislocation was idiopathic in 71 patients (94 knees; 56.0%) and syndromic in 55 (74 knees; 44.0%). The mean age at surgery was 11.5 years (range 4–18) and was significantly lower in syndromic patients. Syndromic patients also exhibited more severe clinical pattern at presentation, based on the Parikh and Lykissas score. The mean follow-up was 5.3 years (range 1.0–15.4). Redislocation occurred in 19 cases, with 10 cases requiring further realignment. The Parikh and Lykissas score and the presence of congenital ligamentous laxity were independent predictors of failure. A total of 22 knees in 18 patients required additional surgical procedures. The post-operative Kujala score was significantly lower in patients with syndromic patellar instability. Conclusions: The type of instability and the presence of underlying syndromes negatively affect the rate of redislocation and the clinical and functional outcome following patellar realignment through the three-in-one procedure. We recommend the consideration of alternative surgical strategies, especially in children with severe syndromic patellar dislocation.

## 1. Introduction

Patellar instability is one of the most common disorders of the knee during childhood and adolescence [1,2]. It includes several different conditions, such as acute dislocations, subluxations, recurrent instability, and congenital dislocations [3,4]. The etiology is multifactorial involving trauma, congenital bone abnormalities, neuromuscular impairment, and ligamentous laxity [5,6].

Patellar dislocation in children and adolescents is more challenging because of the heterogeneity of the clinical patterns at presentation, the skeletal immaturity, and the frequency of associated conditions. Several classification systems have been proposed for enabling surgeons to recommend specific treatment options and for allowing comparison among different conditions and treatments [5,7,8,9,10,11]. Recently, Parikh and Lykissas proposed a comprehensive classification system which classified patella instability into four types based on increasing severity and complexity of required treatment (see Table 1) [4]. While type 1 and type 2 dislocations are commonly seen in adolescents and young adults, type 3 obligatory (habitual) and type 4 congenital (fixed) dislocations are less frequent. In fact, type 3 and 4 dislocations are generally encountered in young children and are the most demanding patterns, due to the potential underlying etiologies and complex anatomic pathology. These dislocations can be associated with neuromuscular diseases (i.e., cerebral palsy), congenital deformities (nail–patella syndrome, amyoplasia, congenital limb defects), and collagen disorders due to chromosomic or genetic pathologies (Down syndrome, Elhers–Danlos syndrome, etc.). Patients with congenital or obligatory dislocation may present with knee malalignment, which may be either primary, contributing to the patellar dislocation mechanism, or secondary to the malpositioned extensor mechanism. Flexion contracture and limping are also typical findings [12].

Although conservative management is generally indicated in acute lesions without osteochondral defects, surgical treatment is recommended in the case of recurrent patellar instability, osteochondral fractures with loose bodies, and failed non-operative measures [13,14,15,16]. There is a consensus that the surgical results may be affected by the type of dislocation and the presence of underlying pathologies [4,13,17]. However, the impact of these factors has not been clearly quantified.

Therefore, the aim of the present study is to report the mid- to long-term results of a cohort of children and adolescents affected by various types of patellar instability, with or without underlying pathologies, that underwent patellar realignment through a three-in-one surgical procedure. Our question is if the type of patellar instability and the associated conditions may affect the post-operative results.

## 2. Materials and Methods

### 2.1. Study Design

The present study is a retrospective analysis of a cohort of skeletally immature patients affected by patellar instability, undergoing patellar realignment through a three-in-one procedure. Our unit is a tertiary referral department for pediatric orthopedics, which treats approximately 20–30 cases of patellar instability per year. The hospital’s electronic medical records (charts and radiograms) were used to collect data retrospectively. Skeletally immature patients with still-open physes who underwent the three-in-one procedure were included in the study. Conservative treatment and other surgical treatments (such as isolated lateral release, MPFL reconstruction, isolated proximal or distal realignment, Langenskiöld or Stanisavljevic techniques) were considered exclusion criteria. The investigation was conducted by independent observers who were not involved in the treatment of the patients.

### 2.2. Baseline Variables

Clinical baseline variables included age, sex, body mass index (BMI) percentile adjusted for sex and age, history of trauma, comorbidities, and previous operations. From an etiologic point of view, the patella dislocation was classified as idiopathic or syndromic. The latter encompasses patients with neuromuscular disorders (e.g., cerebral palsy), connective tissue disorders (e.g., Ehlers–Danlos syndrome), or other syndromes with abnormalities of the lower limb (e.g., longitudinal defects of the lower limb, amyoplasia, etc.).

The pattern of patellar instability was classified according to the Parikh and Lykissas score by two of the authors (G.T. and S.S.) who independently read medical records: Type 1 means first-time patellar dislocation, type 2 means second or subsequent patellar dislocation or continued symptoms after an initial instability episode, type 3 is a habitually dislocatable (actively or passively) patella, and type 4 is a permanently dislocated (reducible or fixed) patella (see Table 1).

The radiographic baseline variables included patellar height evaluated according to the Caton–Deschamps methods, patellar congruence, patellar tilt, Tibial Tuberosity-Trochlear Groove distance (TT-TG), and Hip–Knee–Ankle (HKA) angle [18,19,20,21,22,23].

### 2.3. Surgical Technique

In all the cases, we performed a three-in-one procedure (See Figure 1A–G). The surgical technique was already described elsewhere [24]. Briefly, a long lateral parapatellar incision was used, extending 2 cm from the supero-lateral pole of the patella, to 2–3 cm below the tibial tuberosity. A quadricepsplasty, including an extensive lateral retinacular release (leaving the synovium intact, when possible) and a dissection of the vastus medialis obliquus from the quadriceps tendon was performed, according to Green [25]. Afterward, the vastus medialis complex was transferred distally and laterally to the anterior surface of the patella. The patellar tendon was split longitudinally and the lateral half was detached from its distal insertion and passed medially, beneath its intact medial half, and then sutured in a pouch of periosteum, under the insertion of the pes anserinus [24,26]. The knee was held at about 30° of flexion during the operation, to avoid excessive tension to the patellar tendon.

### 2.4. Post-Operative Management and Follow-Up

The leg was immobilized in an above-the-knee cast with 30° of knee flexion for a 4-week period. Then, a period of 12–20 weeks of physical therapy and the use of a patellar bracing were recommended.

The rate of recurrent instability, the rate of reoperation, and the rate of additional surgical procedures were reported. An Italian validated version of the Kujala Anterior Knee Pain Scale (AKPS) Score [27,28], was administered to each patient at the last evaluation at follow-up or with a self-filled form. For non-cooperating patients (e.g., young children and syndromic patients), parents were asked to answer the questions. The final score was rated as good/excellent (AKPS ≥ 85), fair (AKPS 65–84), and poor (AKPS < 65) [29].

### 2.5. Data Analysis

Patients were assigned a numerical code, and their data were entered into Excel (Microsoft, Redmond, WA, USA) and SPSS (version 22.0; SPSS, Chicago, IL, USA). Continuous data were expressed as mean ± standard deviation (SD) and range, whereas categorical and ordinal data were expressed as raw numbers and proportions with a 95% confidence interval (C.I.). Normality was tested using the χ^2^ test for categorical variables and the Kolmogorov–Smirnov test for continuous variables. Differences between groups were analyzed using the Fisher’s exact test for categorical variables and the Student’s *t*-test (normal data) or the Mann–Whitney U test (skewed data) for continuous variables. The Spearman’s rank test was used for correlations. Exploratory univariate analyses with General Linear Models were performed to assess the impact of the baseline variables on the outcome. Unconditional logistic regression with Wald backward selection was then used to adjust for variables identified in univariate analysis to be significantly different between the outcomes. The survival free from redislocation was calculated using the method of Kaplan and Meyer, and we conducted Cox proportional hazards modeling entering variables found to be significantly related to the survival time free from redislocation. A *p*-value of <0.05 was considered to be statistically significant, and all reported *p*-values were 2-sided.

## 3. Results

### 3.1. Demographics and Clinical/Radiographic Parameters at Baseline

From May 2004 to July 2020, 126 patients (168 knees) underwent the three-in-one procedure for patella realignment. Demographic, clinical, and radiographic baseline data are reported in Table 2.

The patellar dislocation was idiopathic in 71 patients (94 knees; 56.0%) and syndromic in 55 (74 knees; 44.0%). Among the idiopathic patients, a history of trauma was reported in 12 cases. Among the syndromic children, thirty had congenital ligamentous laxity (fourteen Down syndrome, four Elhers–Danlos syndrome, four multiple epiphyseal dysplasia/spondyloepiphyseal dysplasia, two DiGeorge syndrome, one Jacobsen syndrome, one Prune belly syndrome, one Marfan syndrome, one type I osteogenesis imperfecta, one Prader–Willi syndrome, one Coffin–Siris syndrome), eight had neuromuscular disorders (seven cerebral palsy, one muscular dystrophy), and eighteen had congenital deformities of the lower limb (five amyoplasia congenita, two torsional malalignment, two sequelae of tibial and femoral bone infection during infancy, one Sotos syndrome, one Moebius syndrome, one VACTERL syndrome, one nail–patella syndrome, one congenital pseudarthrosis of tibia in NF1, three tibial hemimelia).

As expected, syndromic patients had a lower age at surgery (mean difference = 2.4 years; 95% C.I.: 1.3–3.5 years; *p*-value = 0.0001, see Table 3) and more severe clinical pattern at presentation (*p*-value = 0.0001, see Figure 2 for details).

With the available radiographic imaging, only patellar tilt was significantly different between idiopathic and syndromic patients (see Table 3).

### 3.2. Complications and Rate of Redislocation

A total of 14 patients (20 knees; 11.9%) did not return for follow-up visits and were unavailable to phone or email interview. For the remaining patients, the mean follow-up averaged 5.3 ± 3.1 (1–15.4) years and was significantly higher in syndromic patients (*p*-value = 0.0001). We did not report any infection and/or wound healing problem. Recurrent dislocation was observed in 19 knees (12.8%; 95% C.I.: 7.9–19.3%), with 14 cases (9.5%; 95% C.I.: 5.3–15.3%) receiving further realignment procedures. Patients with redislocation had a lower age at surgery (mean difference 2.0 years, 95% C.I.: 0.2–3.8, *p*-value 0.0303) and higher grade of severity according to the Parikh and Lykissas score (Mann–Whitney U test *p*-value 0.007). In particular, the rate of recurrent instability was 5.2% in type 2 of the Parikh and Lykissas score, 13.2% in type 3, and 24.3% in type 4.

The rate of recurrent instability was also higher in syndromic cases compared with idiopathic cases (19.1% vs. 7.5%, *p*-value = 0.031), especially in children with congenital ligamentous laxity (see Figure 3). However, in a logistic regression model including age at surgery, Parikh and Lykissas score, and congenital ligamentous laxity, children with congenital ligamentous laxity showed a five-fold increased risk for redislocation, compared to the rest of the cohort (OR 5.4, 95% C.I. 2.0–15.0, *p*-value 0.001, see Figure 3).

### 3.3. Survival Free from Redislocation

The overall cumulative 5-year survival rate free from redislocation was 87.0% (95% C.I.: 83.5—90.5%), while the 10-year survival rate was 70.6% (95% C.I.: 63.1—78.1%). The log-rank test confirmed that significant differences existed in the survival curves between children with syndromic congenital ligamentous laxity compared to the rest of the cohort (10-year survival: 51.6% vs. 86.4%; *p* = 0.018, see Figure 4).

With the numbers available, none of the other pre-operative variables significantly affected the survival rate free from redislocation.

### 3.4. Additional Surgical Procedures

A total of 22 knees (13.1%) in 18 patients underwent additional staged surgical procedures (ten femoral and/or tibial osteotomies, three hamstring lengthenings, four femoral medial hemiepiphysiodesis, one limb lengthening, one controlateral femur shortening osteotomy, one subtalar arthroeresis, one fixation of osteochondral fragment, and one open reduction and fixation with K-wire of knee subluxation in DiGeorge syndrome). The proportion of additional surgical procedures was significantly higher in syndromic children (23.0% vs. 5.3%; *p*-value = 0.001, see Figure 5) and correlated with Parikh and Lykissas score (5.6% in grade 2, 10.3% in grade 3, 30.8% in grade 4; *p*-value = 0.001).

### 3.5. Clinical and Functional Outcomes

At the latest follow-up, The AKPS score was available in 136 knees and averaged 86.4 ± 16.3 points (28–100); overall, 85 knees had good/excellent results (62.5%), 38 had fair results (27.9%), and 13 had poor results (9.6%). Syndromic patients had significantly lower scores compared to idiopathic patients (mean difference = 9.1 points; 95% C.I.: 3.7–14.1; *p*-value = 0.001, see Table 4). Among the patients who did not return the AKPS, one patient developed avascular necrosis of the patella, with severe chondral damage, early onset osteoarthritis, and poor outcome. Another patient refused to participate to the survey, since he was leg amputated due to congenital pseudarthrosis of the tibia in NF1.

## 4. Discussion

Many papers in the current literature faced with the management of patellar dislocation in a pediatric population and several surgical techniques have been described to realign the patella in skeletally immature patients [6,30]. Currently, there is evidence that surgical treatment is superior to conservative treatment in reducing the recurrence of the dislocation [13,14,15,16,31]. In particular, the combination of proximal and distal procedures has demonstrated high effectiveness in preventing relapse [14,32]. However, there is still poor evidence about the superiority of any technique of patellar realignment over the others [33].

Pagliazzi et al. reported a meta-analysis of 10 RCTs and 510 patients, demonstrating that surgical treatment significantly reduces the rate of redislocation at short (9% vs. 21%; Risk Ratio for redislocation = 0.40 in favor of surgery) and medium follow-up (21% vs. 30% Risk Ratio for redislocation = 0.58 in favor of surgery) and improves clinical and functional outcomes compared to nonoperative treatment [16]. However, in the subgroup analysis on the adolescent population, the authors noticed a slight increase in the pooled rate of redislocation (23% in the surgical group, 31% in the non-surgical group), concluding that, although trends were similar, the higher redislocation rate in adolescents, together with the variability of the clinical outcome, suggests large heterogeneity of this peculiar population.

Another systematic review including retrospective case series and non-randomized studies reported results from 21 studies and overall 448 knees in pediatric patients treated with various surgical procedures. The authors reported a cumulative rate of redislocation of 13.8%, consistent with our results [34]. The authors also confirmed the large variation of results among studies (from 0% to 82%), while excluding syndromic cases from analysis, and no superiority of a specific technique over the others.

The main issue of these systematic reviews is that the most valuable RCTs and prospective studies were tailored on homogeneous populations, with inconsistent applicability to the broad spectrum of patients with patellar instability [14]. Children with high-grade patellar instability and underlying conditions, such as neuromuscular disorders, collagen disorders, and congenital deformities, were generally excluded from these studies, and only sparse case series dealt with these rare conditions [24,35,36]. Therefore, a gap in knowledge exists about the role of congenital pathologies in affecting the surgical results of these complex cases, that, on the other hand, are not so uncommon in highly specialized referral centers for pediatric orthopedics.

Different from sports medicine and adult knee services, in our experience, children with idiopathic and syndromic patellar dislocation were almost equally represented, since our unit is a tertiary referral center for pediatric orthopedics and rare bone diseases. This peculiarity allows us to deal with the entire spectrum of patellar instability, but it lowers the consistency of our information about prognosis. To stratify these heterogeneous cohorts of patients, comprehensive and reliable classification tools are required. We used a recent classification system proposed by Parikh and Lykissas. Despite the fact that several scores were proposed in the past for classifying patellar instability in children [5,7,8,9,10,11], the system proposed by Parikh and Lykissas seems to be comprehensive, reliable, and simple to adopt, since it was based only on the clinical history and patient examination, not requiring specific manual or instrumental tests. We demonstrated that this classification can accurately predict the risk of redislocation after patellar realignment in children; therefore, we recommend this classification both for research purposes (data standardization) and in clinical practice (for decision making).

We also quantified the risk of recurrent dislocation for children with congenital disorders and syndromes. We found that, compared to idiopathic cases, syndromic patients have more than twice the risk of recurrent dislocation. In particular, children with congenital collagen disorders have a five-fold increased risk of redislocation, and it is conceivable that almost 50% of cases will relapse within ten years after surgery. Other sparse series and case reports regarding recurrent, obligatory, and fixed patella dislocation in syndromic children and adolescents treated with various techniques showed a rate of residual instability and redislocation ranging from 0 to 100% [24,35,37,38,39]. These case series also reported results about different techniques, such as the four-in-one procedure [35,40], the Langenskiöld procedure [41,42], and the Stanisavljevic procedure [37,38,43], which are considered the treatment of choice in congenital obligatory or fixed patellar dislocation. However, due to the rarity of these conditions, no evidence is available about the superiority of each technique over the others.

The use of the three-in-one procedure in skeletally immature patients has been previously reported in the literature. Myers et al. described 42 knees in 37 children undergoing the three-in-one procedure to treat recurrent patella dislocation. Among them, 11 patients had congenital ligamentous laxity [44]. The authors reported a redislocation rate of 9.5% at 2 to 7 years of follow-up, good or excellent clinical outcomes in 76%, and no association between redislocation and congenital ligamentous laxity. They also reported a deterioration of results over time, which is consistent with our findings. In fact, based on the Kaplan–Meyer analysis, we estimated an 86.6% survival rate free from redislocation at 5 years of follow-up, but only 70.9% at 10 years and even lower in the subgroup with congenital ligamentous laxity (51%). Bettuzzi et al. reported results from six children (10 knees) with patella instability and Down syndrome treated by the three-in-one procedure [24]. They reported no redislocation and overall improved symptoms and function at 8.7 years of follow-up. Oliva et al. reported results from a cohort of 25 children with unilateral recurrent patellar dislocation (at least two documented dislocations) undergoing the three-in-one procedure [45]. The authors selected only children without major knee deformities (such as genu valgum, increased Q-angle, patella alta, torsional deformities, patellar and trochlear dysplasia) and did not mention syndromic case management. The authors reported only one case of redislocation (4%) and general improvement of clinical and functional outcomes, although they noticed permanent side-to-side differences in isokinetic strength and thigh diameter, despite subjective success of the procedure. Malecki et al. compared 31 knees undergoing the three-in-one procedure and 32 knees receiving MPFL reconstruction by the adductor magnus tendon [46]. The authors did not report significant differences between groups regarding the rate of redislocation and functional results, although they found a significant difference in pain complaints, favoring the MPFL group.

The need for additional procedures for achieving a stable patella and restoring knee function and alignment is another matter of concern. In our experience, only 13% of patients underwent additional staged procedures for improving knee alignment. This could explain the high number of redislocations, especially in syndromic patients with irreducible patellar dislocation. Although bony procedures, such as trochleoplasty or transposition of the anterior tibial tuberosity, are generally contraindicated in such young patients, the restoration of the knee axis through epiphysiodesis or sparing-physis osteotomies is sometimes required for correcting severe genu valgum, knee flexion, or femoral torsion, especially in children with congenital deformities or neurologic disorders. Alternatively, other surgical procedures, such as the Langenskiöld or the Stanisavljevic procedures, should be considered for these severe conditions. For this purpose, meticulous pre-operative planning is required. The recent advancement in the field of computer-aided surgical simulation, virtual surgical planning, and motion analysis will provide us with innovative digital solutions for increasing the rate of successful and durable long-term outcomes even in extreme cases [47,48].

### Limitations

The strengths of our study include the large number of cases, the use of a comprehensive, reliable, and simple classification of pediatric patellar instability, the homogeneity of surgical treatment, and the heterogeneity of the population, which allowed us to investigate the role of rare conditions as potential risk factors for treatment failure.

However, our study has limitations. First, the retrospective design limits the amount of data that could be obtained by medical records and the significance of the results. For instance, radiographic imaging was insufficient to investigate the role of radiographic parameters as predictors of redislocation and criteria for alternative surgical treatments. Second, the procedures were performed by 13 different surgeons, the indication to surgery and the choice of the procedure was made according to the treating surgeon’s experience and preference, and no established criteria or algorithms for surgical decision making were used. Third, we merged multiple pathologic conditions into single subgroups; this could have introduced selection bias. Moreover, we cannot exclude with absolute certainty that some “idiopathic” patient could not have underlying undiagnosed collagen disorders or other syndromes. Fourth, the Parikh and Lykissas score was applied retrospectively, and its reliability is questionable. In particular, we decided not to consider A and B subtypes in type 3 and type 4 patellar dislocation. However, subtypes could be very different entities of patellar instability with possibly different prognosis. Finally, the Kujala AKPS was obtained only at follow-up, and no comparison with the pre-operative status was possible. This is essential to understand how effective the surgical treatment is in improving symptoms and function, especially in syndromic patients, when the surgical choice may be debatable.

## 5. Conclusions

The three-in-one procedure can be still considered a suitable treatment option for patella realignment in skeletally immature patients. However, the etiology and the type of patellar instability must be strictly considered in surgical decision making, and alternative techniques and strategies must be recommended, as the three-in-one procedure could be ineffective in the most severe patterns of dislocation.

## Figures and Tables

**Figure 1 jcm-12-00702-f001:**
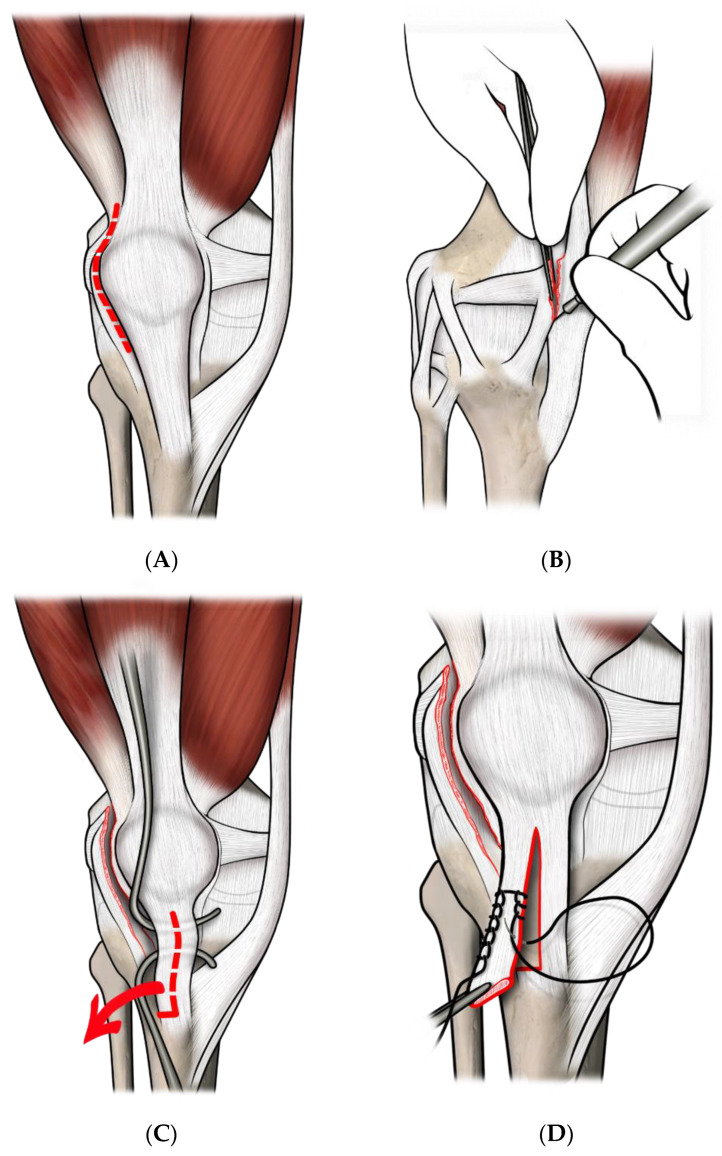
(**A**) lateral parapatellar incision; (**B**) extensive lateral retinacular release; (**C**) the patellar tendon was split longitudinally (red arrow); (**D**) the lateral half was detached from its distal insertion and passed medially; (**E**) the lateral half was sutured in a pouch of periosteum, under the pes anserinus (red arrow); (**F**) the vastus medialis obliquus was detached from the quadriceps tendon, and the medial capsule was dissected; (**G**) the vastus medialis obliquus and the medial capsule were transferred distally and laterally to the anterior surface of the patella (red arrow).

**Figure 2 jcm-12-00702-f002:**
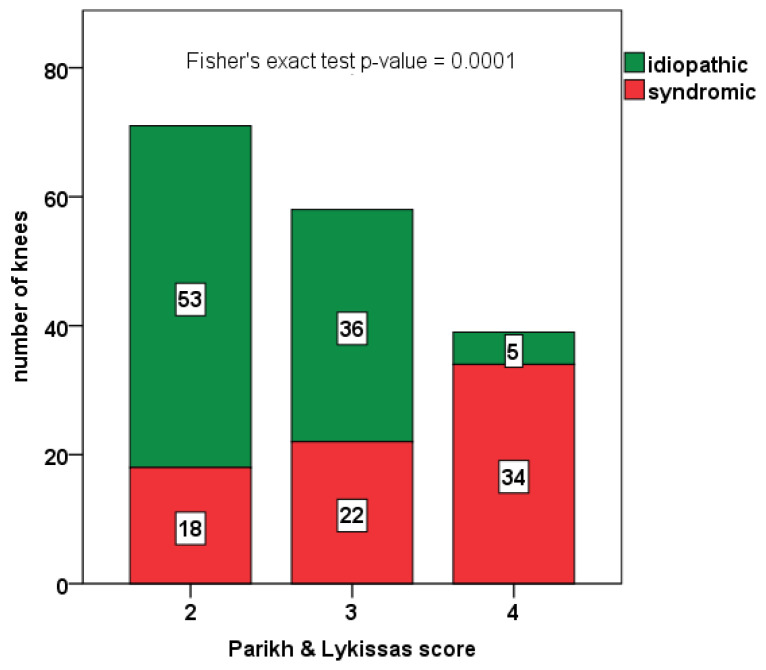
Distribution of idiopathic (depicted in green) and syndromic knees with patellar instability (depicted in red) in groups by Parikh and Lykissas score. Fisher’s exact test *p*-value = 0.0001.

**Figure 3 jcm-12-00702-f003:**
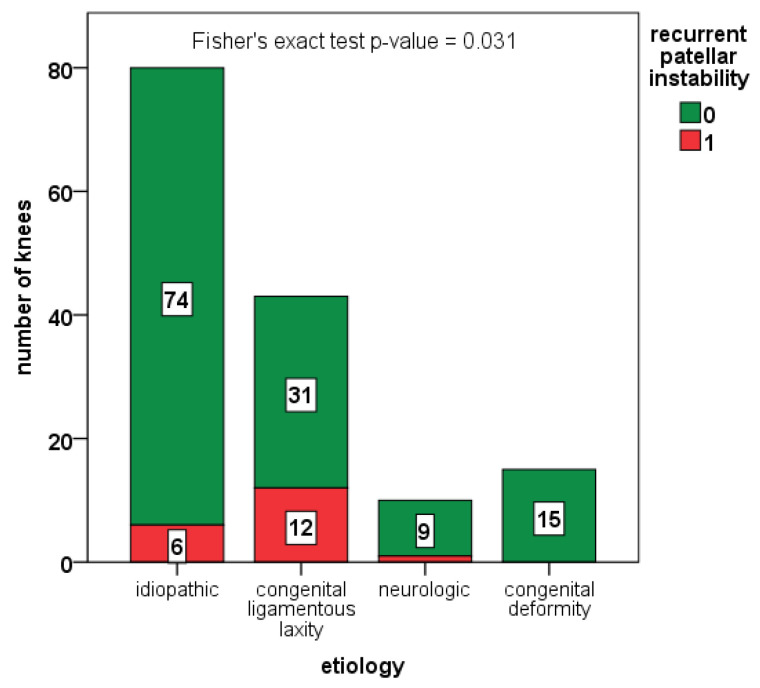
Number of redislocations (depicted in red) among the different subgroups of patients, based on etiology.

**Figure 4 jcm-12-00702-f004:**
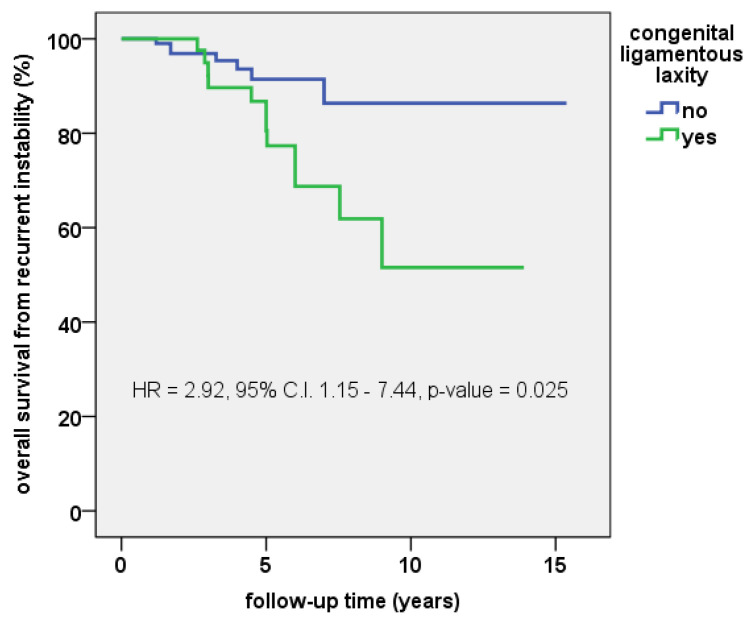
Kaplan–Meyer survival curve free from recurrent instability in children with and without syndromic congenital ligamentous laxity. HR = hazard ratio; C.I. = confidence interval.

**Figure 5 jcm-12-00702-f005:**
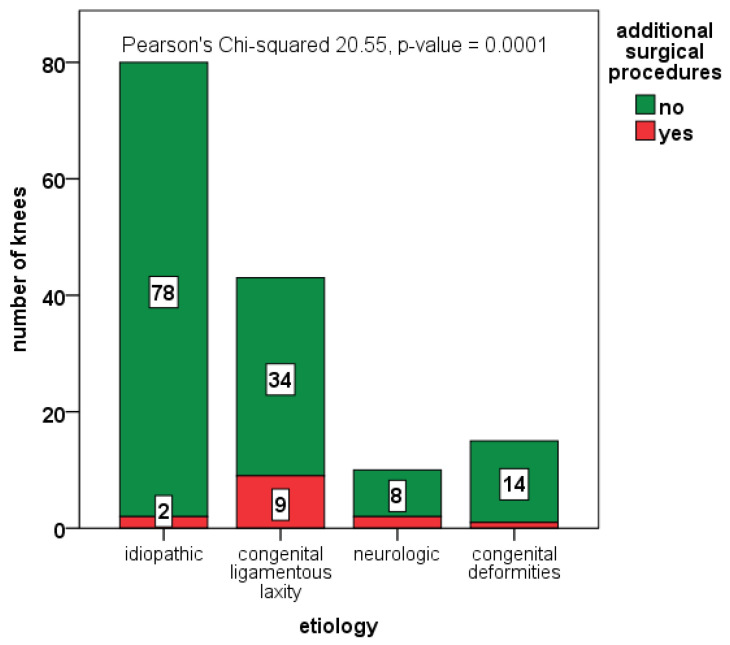
Number of additional surgical procedures (depicted in red) among the different subgroups of patients, based on etiology.

**Table 1 jcm-12-00702-t001:** Classification of patellar instability according to Parikh and Lykissas [4].

Parikh and Lykissas Classification Type/Subtype	Description
Type 1	First patellar dislocation
A	With osteochondral fracture
B	Without osteochondral fracture
Type 2	Recurrent patellar instability
A	Recurrent patellar subluxation
B	Recurrent (>2) patellar dislocation
Type 3	Dislocatable patella
A	Passive patellar dislocation
B	Habitual patellar dislocation in flexion or extension
Type 4	Dislocated patella
A	Reducible
B	Irreducible

**Table 2 jcm-12-00702-t002:** Demographic and radiographic baseline data.

Baseline Variable	Measurement	Value
Patients/Knees	Number (male/females)	126/168 (46/80)
Age at treatment (years)	Mean ± SD (range)	11.5 ± 3.7 (4.1–17.6)
BMI (kg/m^2^)	Mean ± SD (range)	20.2 ± 4.3 (11.2–38.1)
BMI (percentile)	Median (IQR)	74.5 (37.0–93.0)
Caton–Deschamps ratio	Mean ± SD (range)	1.4 ± 0.4 (0.6–2.6)
Congruence angle (°)	Mean ± SD (range)	39.1 ± 32.5 (2.6–128.4)
Patellar tilt (°)	Mean ± SD (range)	26.1 ± 25.5 (0–127.3)
Sulcus angle (°)	Mean ± SD (range)	149.0 ± 15.8 (111–180)
TT-TG distance (mm)	Mean ± SD (range)	14.1 ± 8.1 (5–20.6)
HKA angle	Mean ± SD (range)	7.6 ± 8.5 (−10.4 varus–+41.0 valgus)

SD = standard deviation; BMI = body mass index; TT-TG = tibial tuberosity-trochlear groove; HKA angle = hip–knee–ankle angle; IQR = interquartile range.

**Table 3 jcm-12-00702-t003:** Differences in baseline clinical and radiographic variables between idiopathic and syndromic cases.

Baseline Variable	Group	N	Mean ± SD	*p*-Value
Age at treatment (years)	Idiopathic	94	12.6 ± 3.5	0.0001 *
	Syndromic	74	10.2 ± 3.6	
BMI	Idiopathic	91	21.1 ± 4.4	0.002 *
	Syndromic	67	19.0 ± 4.0	
Caton–Deschamps ratio	Idiopathic	47	1.4 ± 0.6	0.649
	Syndromic	40	1.4 ± 0.4	
Congruence angle (°)	Idiopathic	46	34.89 ± 29.9	0.172
	Syndromic	34	44.9 ± 35.4	
Patellar tilt (°)	Idiopathic	45	18.4 ± 11.7	0.001 *
	Syndromic	33	36.6 ± 34..3	
Sulcus angle (°)	Idiopathic	47	148.7 ± 14.3	0.848
	Syndromic	34	149.4 ± 17.9	
TT-TG distance (mm)	Idiopathic	20	12.4 ± 5.5	0.884
	Syndromic	13	12.7 ± 6.5	
HKA angle (°)	Idiopathic	33	7.5 ± 7.0	0.943
	Syndromic	36	7.7 ± 9.8	

HKA angle = hip–knee–ankle angle; N = number of knees; SD = standard deviation. *: difference between groups were statistically significant (*p*-value < 0.05).

**Table 4 jcm-12-00702-t004:** Post-operative variables.

Post-Operative Variable	Group	N	Mean ± SD	*p*-Value
Follow-up time (years)	Idiopathic	79	4.2 ± 2.6	0.0001 *
	Syndromic	68	6.5 ± 3.3	
Kujala AKPS score	Idiopathic	74	90.5 ± 13.1	0.001 *
	Syndromic	62	81.4 ± 18.3	

N = number of knees; SD = standard deviation; AKPS = anterior knee pain scale. *: difference between groups were statistically significant (*p*-value < 0.05).

## Data Availability

The datasets generated during and/or analyzed during the current study are available from the corresponding author on reasonable request.

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
