# Peer review of "A 20-Year Retrospective Study of Children and Adolescents Treated by the Three-in-One Procedure for Patellar Realignment"

_jcm, 2023, doi:10.3390/jcm12020702_

Round 1
Reviewer 1 Report
I applaud the authors for putting a fair amount of work into this paper which is a retrospective review of a large cohort of Skeletally immature patients that had varying types of patellar instability.
That said, I believe that the vast difference in the patient cohort leads the conclusions that are not helpful to the clinician.
In regards to their patient's cohort with what they term as "idiopathic" or type two instability, their series adds nothing to the literature. It is not a modern-day technique and should not be advocated for all skeletally immature patients.
The distal medial advancement of the VMO is no longer felt to be appropriate for proximal stabilization and has been largely replaced by an MPFL reconstruction.
The years the cohort was derived were 2004 to July 2020. I am surprised that this procedure continued to be used throughout this time period with no change in their approach over this 16-year span of time. With so much advancement in patellofemoral thinking and its surgical algorithm, this is disappointing. Even their post-op regiment of immobilizing the knee in a cast for four weeks is also not common practice, especially in those with what they term idiopathic dislocations.
In regards to their distal realignment, they performed a type of Roux-Goldthwaithe procedure which medializes the lateral half of the patella tendon. This continues to be within the modern algorithm of those that have open growth plates and a high Q vector. However, according to their table 2 reporting on radiographic baseline data, the TTTG distance had a mean of 12.6 mm in a range of 0 to 24 mm. Clearly a medialization of the patella tendon because of a high Q vector was not an appropriate surgery to perform in all of these patients, as their mean is a 'normal' value and it is not known how many were 'high'. Nor is it mentioned how their measurements were made, in particular, CT s MRI.
Further discussion of table 2 on Demographic and radiographic baseline data, there is much that could be questioned.
Their age range was 4 to 19.4 years old, was this at baseline or a final follow-up? It should be baseline and it is surprising that a 19-year-old would still have open growth plates.
For their BMI the value was given as a percentile which is confusing to me and it’s not understandable nor the common lingo that I am used to seeing BMI represented. This is perhaps appropriate per due to the varying age range but an absolute BMI might have also be helpful.
For their C/D ratio or their measurement of the tower height, their range is from .6 to 4.8. I’m not quite sure how one gets a C/D index of 4.8
For their HKA angle, their mean was 7.4 with a range of -10 to 41. I am going to assume that the negative value represents the varus alignment but this should be stated. The value of 41 seems nearly impossible for me to comprehend. However, if it is true then why was not some attention paid to re-aligning the limb at the same time of patella stabilization
I also don’t see anywhere where they’ve divided the number of patients that they had in each category. In lines 154 to 165, they do give the diagnoses of the syndromic category, which has a multitude of different neuromuscular disorders and syndromic problems. This offers its own set of complexities.
But more importantly, we do not know how many are in the categories that they denote as a syndrome and which would be their type three A and B and their type four A and B.
Currently, their categorization of type 3B which would be a habitual patella dislocation in flexion or extension covers two very different patellar entities that being those with a profound J sign (dislocation in extension) versus type 3 B which have an obligate dislocation in flexion.
These patients need totally different surgeries in my opinion. Likewise including type four B, which would be an irreducible wholly dislocated patella, both of these categories might need a quadricep lengthening procedure.
My point is that to approach all of these kinds of types of patients with the same surgery is not something that I would advocate in 2020. If this was a historic review that would be one thing but to advocate for this as a technique that is usable in 2022 would not have anything to do with evidence-based medicine in 2022.
Author Response
I applaud the authors for putting a fair amount of work into this paper which is a retrospective review of a large cohort of Skeletally immature patients that had varying types of patellar instability.
That said, I believe that the vast difference in the patient cohort leads the conclusions that are not helpful to the clinician.
We disagree with the reviewer, because there are only few studies that tried to quantify the impact of comorbidities in the surgical outcome after patellar realignment. The estimation of the risk of redislocation in syndromic patients may be helpful from a clinical point of view, since it allows to better define the surgical decision making and inform parents about the actual risk of failure after surgery.
In regards to their patient's cohort with what they term as "idiopathic" or type two instability, their series adds nothing to the literature. It is not a modern-day technique and should not be advocated for all skeletally immature patients.
The distal medial advancement of the VMO is no longer felt to be appropriate for proximal stabilization and has been largely replaced by an MPFL reconstruction.
The years the cohort was derived were 2004 to July 2020. I am surprised that this procedure continued to be used throughout this time period with no change in their approach over this 16-year span of time. With so much advancement in patellofemoral thinking and its surgical algorithm, this is disappointing. Even their post-op regiment of immobilizing the knee in a cast for four weeks is also not common practice, especially in those with what they term idiopathic dislocations.
We thank the reviewer for this comment and we agree that a modern approach to the patellar instability should consider MPFL repair or reconstruction. In our Unit we begun to perform MPFL reconstruction in 2016, however most of surgeons in our Unit still continue to perform VMO advancements. In the Literature only few studies compared medial retinaculum plasty with MPFL reconstruction, finding no difference in recurrent instability rate (Ma LF Arthroscopy 2013, doi: 10.1016/j.arthro.2013.01.030, Malecki K Int Orthop 2016, doi: 10.1007/s00264-016-3119-1). Two recent reviews affirmed that there is still scarce evidence concerning the superiority of MPFL reconstruction over medial retinaculum plasty and other types of repair (Wilkens OE KSSTA 2020, doi: 10.1007/s00167-019-05656-3; Murray IR 2022 Clin Sports Med, doi: 10.1016/j.csm.2021.07.006). We added a further comment in the discussion.
In regards to their distal realignment, they performed a type of Roux-Goldthwaithe procedure which medializes the lateral half of the patella tendon. This continues to be within the modern algorithm of those that have open growth plates and a high Q vector. However, according to their table 2 reporting on radiographic baseline data, the TTTG distance had a mean of 12.6 mm in a range of 0 to 24 mm. Clearly a medialization of the patella tendon because of a high Q vector was not an appropriate surgery to perform in all of these patients, as their mean is a 'normal' value and it is not known how many were 'high'. Nor is it mentioned how their measurements were made, in particular, CT s MRI.
We thank the reviewer for this comment. The TTTG distance was measured in absolute values in CT and/or MRI, when available. In pediatric population this measure changes with chronologic age. Our cohort averaged between 10 and 12 years old, in which median TTTG reported in Literature is 8.5 mm and a value of more than 12.1 mm is considered abnormal (see Dickens AJ et al. Am JBJS 2014).
Further discussion of table 2 on Demographic and radiographic baseline data, there is much that could be questioned.
Their age range was 4 to 19.4 years old, was this at baseline or a final follow-up? It should be baseline and it is surprising that a 19-year-old would still have open growth plates.
We are grateful for this observation. We reviewed data and excluded the 3 oldest patients with closed growth plates.
For their BMI the value was given as a percentile which is confusing to me and it’s not understandable nor the common lingo that I am used to seeing BMI represented. This is perhaps appropriate per due to the varying age range but an absolute BMI might have also be helpful.
We have data of absolute BMI, but it is not recommended to be used in children as absolute. Since percentile is not a linear value, analysis on BMI was made dividing patients by range groups of BMI percentile. However, we added absolute BMI in Table 2 as requested. We did not observe any correlation between BMI and rate of failure in this cohort of children.
For their C/D ratio or their measurement of the tower height, their range is from .6 to 4.8. I’m not quite sure how one gets a C/D index of 4.8
We thank the reviewer for this observation. We checked the surgery documentation and we found it was mistakenly included in the cohort, since it underwent a patellar lowering procedure, so we removed him from the cohort.
For their HKA angle, their mean was 7.4 with a range of -10 to 41. I am going to assume that the negative value represents the varus alignment but this should be stated. The value of 41 seems nearly impossible for me to comprehend. However, if it is true then why was not some attention paid to re-aligning the limb at the same time of patella stabilization
The patient had a Di George syndrome with bilateral instability and one of the knees had severe valgus deformity, which required also corrective osteotomy. We specified in the text.
I also don’t see anywhere where they’ve divided the number of patients that they had in each category. In lines 154 to 165, they do give the diagnoses of the syndromic category, which has a multitude of different neuromuscular disorders and syndromic problems. This offers its own set of complexities.
We agree with the reviewer, and already specified this in the limitations section. However, other studies already merged syndromic patients in a similar way. We are aware that this is a potential bias.
But more importantly, we do not know how many are in the categories that they denote as a syndrome and which would be their type three A and B and their type four A and B.
Currently, their categorization of type 3B which would be a habitual patella dislocation in flexion or extension covers two very different patellar entities that being those with a profound J sign (dislocation in extension) versus type 3 B which have an obligate dislocation in flexion.
These patients need totally different surgeries in my opinion. Likewise including type four B, which would be an irreducible wholly dislocated patella, both of these categories might need a quadricep lengthening procedure.
The reviewer is right, this is related to the retrospective nature of the study and the retrospective application of Parikh and Lykissas score, that limits the reliability of our findings. We already specified this aspect in the Limitations of the study, but we further underlined this bias in the revision of the manuscript.
My point is that to approach all of these kinds of types of patients with the same surgery is not something that I would advocate in 2020. If this was a historic review that would be one thing but to advocate for this as a technique that is usable in 2022 would not have anything to do with evidence-based medicine in 2022.
We agree with the reviewer, however the aim of the study is not to affirm that the three-in-one procedure is superior in comparison with other techniques, but to report the long term results in wide cohort of patients. We already discussed that the three-in-one procedure may produce unfavorable outcomes in the most complex cases and that accessory procedure or different techniques should be considered in the decision-making process.
Reviewer 2 Report
Nice article. Need some more clarification. The study included all the patients with patellar instability, maybe during investigation time were other patients operated in different way?
Who evaluated patellar instability? The same person or or data collected from medical records?
Could authors stress about surgical complications after surgery? No wound healing problems???
When were patients evaluated post op. The same day? How syndromic patients were evaluated with Kujala Anterior Knee Pain Scale?
Author Response
Nice article. Need some more clarification. The study included all the patients with patellar instability, maybe during investigation time were other patients operated in different way?
Who evaluated patellar instability? The same person or or data collected from medical records?
We specified as requested: the type of patellar instability was evaluated retrospectively from medical records. The Parikh and Lykissas score was assigned by two of the authors.
Could authors stress about surgical complications after surgery? No wound healing problems???
We thank the reviewer for this suggestion. We did not report any infection and/or wound healing problem. We specified it in the Results.
When were patients evaluated post op. The same day? How syndromic patients were evaluated with Kujala Anterior Knee Pain Scale?
We specified as requested: scores were administered with a self-filled form at the most recent follow-up. For non-cooperating patients (young children and syndromic), parents were asked to answer the questions.
Reviewer 3 Report
Dear Authors: I want to congratulate with You for Your work, and for the paper. I suggest very minor revision, with regards to lines 108-110: this is a modified Roux-Goldthwait procedure, You should cite it by name and by article (as You did for Green in line 106).
best regards, and congrats for the huge work.
Author Response
Dear Authors: I want to congratulate with You for Your work, and for the paper. I suggest very minor revision, with regards to lines 108-110: this is a modified Roux-Goldthwait procedure, You should cite it by name and by article (as You did for Green in line 106).
We inserted citations as requested.
Thank you for your time and your suggestion.
best regards, and congrats for the huge work.